# Emerging Microfluidic Tools for Simultaneous Exosomes and Cargo Biosensing in Liquid Biopsy: New Integrated Miniaturized FFF-Assisted Approach for Colon Cancer Diagnosis

**DOI:** 10.3390/s23239432

**Published:** 2023-11-27

**Authors:** Valentina Marassi, Stefano Giordani, Anna Placci, Angela Punzo, Cristiana Caliceti, Andrea Zattoni, Pierluigi Reschiglian, Barbara Roda, Aldo Roda

**Affiliations:** 1Department of Chemistry “G. Ciamician”, University of Bologna, 40126 Bologna, Italy; valentina.marassi@unibo.it (V.M.); stefano.giordani7@unibo.it (S.G.); anna.placci2@studio.unibo.it (A.P.); andrea.zattoni@unibo.it (A.Z.); pierluigi.reschiglian@unibo.it (P.R.); 2National Institute of Biostructure and Biosystems (INBB), 00136 Rome, Italy; angela.punzo2@unibo.it (A.P.); cristiana.caliceti@unibo.it (C.C.); 3byFlow srl, 40129 Bologna, Italy; 4Department of Biomedical and Neuromotor Sciences, University of Bologna, 40138 Bologna, Italy; 5Interdepartmental Centre for Renewable Sources, Environment, Sea and Energy—CIRI FRAME, University of Bologna, 40131 Bologna, Italy; 6Interdepartmental Centre for Industrial Agrofood Research—CIRI Agrofood, University of Bologna, 47521 Cesena, Italy

**Keywords:** liquid biopsy, exosomes, colon cancer, microfluidic, reagent less biosensors, hollow-fiber field-flow fractionation, miniaturization, luminescence

## Abstract

The early-stage diagnosis of cancer is a crucial clinical need. The inadequacies of surgery tissue biopsy have prompted a transition to a less invasive profiling of molecular biomarkers from biofluids, known as liquid biopsy. Exosomes are phospholipid bilayer vesicles present in many biofluids with a biologically active cargo, being responsible for cell-to-cell communication in biological systems. An increase in their excretion and changes in their cargo are potential diagnostic biomarkers for an array of diseases, including cancer, and they constitute a promising analyte for liquid biopsy. The number of exosomes released, the morphological properties, the membrane composition, and their content are highly related to the physiological and pathological states. The main analytical challenge to establishing liquid biopsy in clinical practice is the development of biosensors able to detect intact exosomes concentration and simultaneously analyze specific membrane biomarkers and those contained in their cargo. Before analysis, exosomes also need to be isolated from biological fluids. Microfluidic systems can address several issues present in conventional methods (i.e., ultracentrifugation, size-exclusion chromatography, ultrafiltration, and immunoaffinity capture), which are time-consuming and require a relatively high amount of sample; in addition, they can be easily integrated with biosensing systems. A critical review of emerging microfluidic-based devices for integrated biosensing approaches and following the major analytical need for accurate diagnostics is presented here. The design of a new miniaturized biosensing system is also reported. A device based on hollow-fiber flow field-flow fractionation followed by luminescence-based immunoassay is applied to isolate intact exosomes and characterize their cargo as a proof of concept for colon cancer diagnosis.

## 1. Introduction

Cancer is one of the most common causes of death worldwide, and its early, accurate diagnosis is mainly based on tissue biopsy followed by different imaging techniques. This approach presents some disadvantages, since it is quite invasive and is not fully representative of the tumor’s heterogeneous distribution, increasing the risk of inaccurate diagnosis and metastasis. Emerging non-invasive methods, such as liquid biopsy, represent interesting alternatives by which to diagnose and monitor cancers and collect useful information regarding the overall state of the patient [1,2,3].

Liquid biopsies consist of the detection of tumor-derived markers (e.g., circulating cells or DNA, extracellular vesicles) present in the body fluids of patients, followed by analyses of their genomic and proteomic profiles. Due to its minimal invasiveness, liquid biopsy may drastically improve the field of clinical oncology, permitting continuous monitoring by repeated sampling phases and improving personalized therapeutic approaches [4].

Exosomes are lipid bilayer vesicles released in body fluids, and which play a central role in intercellular communication through the transfer of bioactive molecules (proteins, lipids, and nucleic acid) [5]. Several studies have investigated the role of exosomes both in physiological and pathological conditions, underlining their involvement in the different stages of cancer [6].

Our knowledge of exosomes has rapidly increased during the last 20 years of research, helping to realize their potential role in the diagnosis and therapy of many diseases. It is now well recognized that cancer cells are characterized by high heterogeneity and different subtypes; thus, their identification cannot be based on unique targets. Moreover, the growth of a tumor may be influenced by its own microenvironment as well as the host organism [7]. Exosomes transfer their contents to the recipient cells, thus playing critical roles in tumor progression and allowing the horizontal transfer of information through their “cargo”, made of functional proteins and nucleic acids. Thus, exosomes isolated from biological fluids may be very interesting candidates as biomarkers and/or targets [8,9], providing new opportunities for medical applications, including cancer liquid biopsy (Figure 1) [10,11].

The first exosome-based liquid biopsy test to receive a Breakthrough Device Designation by the FDA was the ExoDx Prostate IntelliScore test, a non-invasive tool for the diagnosis of high-grade prostate cancer using urine samples [12]. The established clinical use of exosomes for liquid biopsy requires appropriate techniques pertaining to their isolation from biological fluids with high efficiency and purity for their analysis, representing a big challenge due to their heterogeneity [13,14].

Current techniques for exosome isolation have numerous weaknesses, being complex, low selective, time-consuming, and unable to yield high purity. The presence of contaminating proteins and RNAs in exosome preparations has been reported; thus, the purity of isolated exosomes is still a key concern [15]. Integrated microfluidic-based biosensors, which are able to directly isolate and detect intact exosomes and simultaneously analyze specific biomarkers contained in their cargo, may represent a solution, with their advantages related to their only requiring a low sample amount, their ability to entrap even very low concentrations of analytes, high-throughput, and being able to offer a rapid and sensitive diagnosis.

In this review, emerging microfluidic tools and biosensing approaches are reported and critically discussed, with further emphasis on the more versatile approaches as well as those based on noninvasive label-free formats. In this regard, we present the design of a new biosensor based on miniaturized flow field-flow fractionation (FFF). As proof of principle, the application for the direct and specific isolation of exosomes from very low volumes of serum from colorectal cancer patients is shown.

## 2. Exosomes for the Early Diagnosis of Cancer

### 2.1. Exosome Biogenesis and Biophysics

Biological fluids or in-vitro-grown cell lines contain extracellular vesicles (EVs). EVs are cell-secreted natural carrier systems that can transfer nucleic acids, proteins, and lipids between donor and recipient cells. Based on their size and mechanism of release, three types of EVs can be distinguished: exosomes (30–200 nm) that are multivesicular body-derived bilayer membrane vesicles with a density of 1.13–1.19 g/mL, microvesicles (<1000 nm) arising from the budding of the plasma membrane, and apoptotic bodies (>1000 nm) deriving from blebbing of the apoptotic cell membrane. The mechanism for exosome generation and release goes through the formation of multivesicular bodies (MBs) that encompass the exosomes in the cytoplasm (Figure 2A) [16]. During this process, biomolecules are incorporated into the invaginating membrane, while the cytosolic elements are held within. MBs biogenesis is controlled by multiple mechanisms, mainly based on the lipid microdomain present in the plasma membrane and the cytosolic protein complexes (endosomal sorting complex required for transport). Then, MBs can fuse with the plasma membrane leading to the secretion of exosomes into the extracellular matrix, or they can undertake a degradation process by fusion with lysozyme. Exosomes can protect carried contents from the mononuclear phagocyte system. It has been described that several complex pathways are able to activate exosome generation and the composition of the exosomes varies depending on the type and physiological state of the cell of origin [17].

Exosomes are involved in many steps of cancer progression, from growth to drug resistance. A large number of exosomes are released from cancer cells, suggesting a preferred exosome secretion as a path for MBs evolution in these cells, where they cause the transformation of healthy cells into cancerous cells, and their release to the extracellular matrix with metastases formation [18]. In cancer cells, the biogenesis of exosomes is regulated by several factors: aberrant gene expression including microRNA, posttranslational modifications, and altered signaling pathways. In addition, hypoxia, decreased pH, and a high concentration of lactate were shown to enhance exosome secretion [19]. The great variety of mechanisms for exosome biogenesis leads to a high heterogeneity of exosome composition. Thus, the simultaneous detection of intact exosomes and their cargo represents a promising option for highly specific cancer diagnosis with perspectives in personalized medicine.

Chemical composition, surface, size, and mechanical features are different among exosome subpopulations according to the cellular origin and physiopathological state; and they contribute to exosome biodistribution and functions in vivo [20]. Exosomes released by cells in physiological and pathological conditions express different membrane proteins, playing key roles in these processes [21]. Regarding molecular composition, exosomes carry many biologically functional macromolecules. Protein composition is related to the endosomal origin (e.g., tetraspanins, Tsg101 and Alix), non-specific proteins such as membrane fusion and transferring proteins (e.g., annexins, Rab and flotillins), heat shock proteins (e.g., Hsc70 and Hsc90), and cytoskeleton proteins (e.g., myosin, actin, and tubulin); and a wide range of cell-type-specific proteins, which can vary dependent on the pathophysiological conditions. Exosomes are enriched in specific lipids involved in the maintenance of exosome morphology, biogenesis, and the regulation of homeostasis in recipient cells. More recently, it has been described that exosomes contain nucleic acids that can be transferred to recipient cells [22] (Figure 2B).

Studies applied to biological fluids highlight the high heterogenicity in the lamellarity, size, and morphology of the exosomes, which results in them being either spherical or tubular with a bilayer membrane of about 5 to 8 nm in thickness [23]. Being colloidal nanoparticles, the physical properties of exosomes can be described through their zeta potential, which represents a measure of charge stability and affects all particle-particle interactions [24,25]. The zeta potential of the exosome and the pH and ionic strength of the biological fluid contribute to the stability and ability of exosomes to properly deliver biomolecules. For example, the surface charge is known to influence different biological functions, such as cellular uptake and cytotoxicity [26]. Generally, a higher zeta potential results in greater electrostatic repulsions between particles with a consequentially low tendency to aggregate. The surface of exosomes will generally be negatively charged due to the nature of molecules expressed at their surfaces. However, different body fluid tissues or cell cultures present large differences in terms of the size and zeta potential values of released exosomes. A lower value of zeta potential was shown for vesicles isolated from the plasma of cancer patients compared to healthy controls [27,28] (Figure 2C).

Thus, the physical and biological properties of exosomes, including their size, surface charge and density, cargo, and membrane-associated antigens, can be exploited for their isolation and characterization. The advanced characterization of the molecular composition associated with each subset of exosomes can facilitate the identification of potential diagnostic/prognostic biomarkers for different pathologies, including cancer.

### 2.2. Exosomes for the Early Diagnosis of Colorectal Cancer

Colorectal cancer (CRC) is one of the four most common solid tumors in the world and ranks second among the causes of tumor death worldwide, with an estimated 0.9 million deaths in 2020 [29]. By 2030, CRC’s global burden will rise to 60% according to demographic studies previsions. Patients with distant metastasis related to CRC have a very low survival percentage (10%) and approximately 25% of the diagnosed patients already show progressing metastasis [30,31].

Colorectal cancer is a complex disease involving multiple genetic, epigenetic, and proteomic changes, and the establishment of its molecular signature is still under debate. Among the genetic changes, the most important are: APC (Adenomatous Polyposis Coli) mutations, which occur in the early stages of tumorigenesis and lead to the dysregulation of the Wnt/β-catenin signaling pathway; KRAS, NRAS, and BRAF mutations, which in turn activate the MAPK signaling pathway, promoting cell proliferation and survival; TP53 (Tumor Protein p53) mutations; and PIK3CA (Phosphatidylinositol-4,5-Bisphosphate 3-Kinase Catalytic Subunit Alpha) mutations, which activate the PI3K/Akt/mTOR pathway, promoting cell survival and growth. TGF-β pathway alterations and the overexpression and activation of EGFR (Epidermal Growth Factor Receptor) signaling [32].

Epigenetic changes are involved in CRC development and progression, such as DNA methylation, altered histone acetylation, methylation, and miRNA dysregulation. The dysfunctional regulation of cell cycle checkpoints, cyclins, and cyclin-dependent kinases (CDKs) have also been observed in CRC cells, leading to uncontrolled cell division and tumor growth. Finally, the tumor microenvironment interacts with stromal cells, immune cells, and the extracellular matrix, which influences tumor growth, invasion, angiogenesis, and metastasis. As previously mentioned, although much progress has been made in discovering the mechanisms underlying colorectal cancer, no readily available circulating biomarkers (i.e., in liquid biopsies, urine, and saliva) have been found yet, leaving the need for early diagnosis still unmet.

Accumulating evidence has reported that CRC initiation and progression are strongly correlated with molecules in tumor-cell-derived exosomes, such as microRNA (miRNAs), long non-coding RNAs (lncRNAs), and proteins, which also provide information about the donor cells’ origin through their content [31,33,34]. Moreover, cells from CRC produce more exosomes than non-cancer cells both in vitro and in vivo, modifying local and distant surroundings and, consequently, concurring tumor development and progression [35]. For these reasons, the development of a fast and efficient method for the isolation of exosomes represents an effective tool to allow the characterization of exosome-derived biomolecules for the early detection and identification of targeted therapies for CRC. The specific molecular signature of exosomes from CRC cells can vary based on the cancer stage, aggressiveness, genetic mutations, and microenvironment. Several exosome-derived biomolecules are currently under investigation: proteins, nucleic acids (RNA and DNA), lipids (phosphatidylserine, phosphatidylcholine, cholesterol, and sphingomyelin), and other metabolites, such as glycans or glycosylated molecules [36,37,38]. Recently, Ogata et al. found seven miRNAs (miR-23a, miR-1246, let-7a, miR-1229, miR-150, miR-223, and miR-21), which are significantly over-expressed in exosomes from serum of patients with CRC at various stages while being undetectable in healthy controls. Furthermore, these miRNA levels significantly decreased after surgical resection, highlighting the existing link between exosomes and tumorigenesis [39]. More recently, Wang et al. reported that the specific miR-125a-3p was significantly over-expressed in plasma exosomes from patients with early-stage CRC, suggesting its possible application as a diagnosis biomarker [40]. Other exosome-derived miRNAs, such as miR-17-92a, miR-92, miR-638, and miR-19a, have been related to CRC as negative prognostic factors; indeed, the elevated serum levels of these miRNAs were variably correlated with lymphatic/vascular infiltration or short relapse-free survival, thus representing possible candidates for recognizing patients at high risk of recurrence after tumor resection [41]. In the era of multi-omic sciences, system biology approaches provided new perspectives from which to see through the complexity of exosomes identifying the biomolecules of different structures. Focusing on lncRNAs, Deng and coworkers reported that lncRNA 91H was abnormally overexpressed in several human tumor tissues and was considerably associated with a worse prognosis in CRC patients [42], at least in part through modulating the heterogeneous nuclear ribonucleoprotein K (HNRNPK) expression and chemoresistance. Similarly, lncRNA RPPH1 was significantly overexpressed in CRC tissues, and the RPPH1 upregulation was related to a poor prognosis and an advanced tumor/node/metastasis (TNM) stage [43]. Dong et al. proved also that lncRNA breast cancer anti-estrogen resistance 4 (BCAR4), mRNA keratin-associated protein 5-4 (KRTAP5-4), and mRNA melanoma antigen family A3 (MAGEA3) were overexpressed in the serum exosomes of CRC patients [44].

Exosomes are also enriched in protein that can influence the behavior of recipient cells, activating signaling pathways involved in cell proliferation and survival. The major proteins associated with CRC progression, cell signaling, and metastasis are tetraspanins (CD9, CD63, CD81) [45], heat shock proteins (HSP70, HSP90) [46], tumor-associated antigens (CEA, CA19-9) [47], and proteins involved in cell adhesion, migration, and invasion (integrins, metalloproteases) [48]. A proteomics analysis reported that 36 proteins were upregulated in the serum exosomes of CRC patients (e.g., alpha-1-antitrypsin (SERPINA1), alpha-2-antiplasmin (SERPINF2), and complement C9 (C9)), while 22 were downregulated, such as the integrin-mediated cell adhesion pathway, fibroblast growth factor receptor 1 (FGFR1), insulin-like growth factor 1 (IGF1), vitronectin (VTN), and chaperonin heat shock protein 90 (Hsp90) [49]. These overexpressed proteins are involved in processes that modulate the microenvironment of metastasis, such as inflammation. In another study, Campanella et al. reported that the amount of chaperonin heat-shock protein-60 (Hsp60) present in the exosomes of enrolled CRC patients was different before and after surgery [50]. This observation is in line with a previous study, where the amount of Hsp60 in the exosomes of patients before surgery was found to be significantly higher than in the exosomes of the same patients after surgery, among which Hsp60 was decreased to levels comparable to those of the controls [51].

Other exosomal proteins already involved in different cancers have been identified as prognostic biomarkers for CRC, including ALIX (ALG 2-interacting protein X), HSP70, CEA, ribosomal protein L13a (RPL13A), hydroxymethylbilane synthase (HMBS), TATA box binding protein (TBP), ATP-binding cassette transporter G1 (ABCG1), copine III (CPNE3), Np73, and Wnt [52]. Moreover, recent data have shown that exosomes may transfer the well-known CRC biomarker mutant KRAS to cells that only produce wild-type KRAS, [53], and exosomes isolated from cells with mutant KRAS significantly increase the cancer-related molecules, including inflammatory cytokines, such as interleukin 8 (IL-8) [54] and IL 6 [55], encouraging neutrophil recruitment, and hence inflammation in the tumor microenvironment, as well as adjacent cells in CRC.

Regarding the current state of the art, exosomes may represent a promising source of biomarkers for CRC diagnosis and for noninvasive detection within biological fluids (i.e., serum, urine, saliva, and feces), thus representing a promising attractive target to investigate in the incoming liquid biopsy era. Indeed, colonic exosomes secreted by epithelial cells are present in the stool thanks to their lipid bilayer structure, which protects them from degradation [56,57], allowing for the detection of their contents such as miRNAs, proteins, and metabolites for CRC diagnosis. However, even if colonic exosomes for the prediction of CRC were successfully isolated and characterized from different biological matrices, their complex extraction from feces remains an obstacle [58]. Very few studies have reported fecal exosome extraction and isolation from human samples and none of them are about CRC investigation [57,59,60]. The analysis of fecal exosomes from CRC patients could represent an innovative field in the future by which to identify cancer-associated molecules as biomarkers for the screening, diagnosis, and, possibly, the treatment of CRC.

## 3. Integrated Microfluidic System for Exosomes Analysis

As mentioned, due to their broad size range, as well as varied surface composition, exosomes are challenging to isolate, quantify, and analyze from biological fluids, though various properties can be exploited for their purification [61]. Exosome isolation from biological fluids remains challenging since each body fluid has its own composition and biophysical properties. In the case of blood, exosomes have to be isolated from serum or plasma, which are highly viscous and very concentrated in proteins, hindering the isolation of pure exosomes [62]. Several pre-analytical factors, such as blood anticoagulant treatment, blood transportation, and storage conditions before isolation, should be taken into account since they can influence the content of exosomes. Moreover, blood is rich in lipoproteins, a significant contaminant, since they are similar in size [63]. After purification, exosomes have to be quantified and characterized. The biochemical content can be determined with proteomics, lipidomics, and Western blotting analysis, which are often limited by exosome heterogeneity [64]. Single-particle analysis, such as atomic force microscopy and electron microscopy, may contribute to morphological characterization; however, these techniques require extensive labeling, are limited in terms of sample throughput, and are only able to assess a small portion of the sample, thus not being representative of the overall sample [65]. Nanoparticle tracking analysis (NTA), tunable resistive pulse sensing (TRPS), and dynamic light scattering (DLS) are often used in size analysis, with limits related to the heterogeneity of nanoparticles [66,67,68]. Hence, the validation of analytical tools for exosome isolation and membrane and biomarkers-cargo analysis is still an open issue.

Recently, the rapid development of technology has prompted the diffusion of microfluidic technologies also integrated into single chips, providing the highly efficient isolation, enrichment, and multi-parametric detection of exosomes. The application of these methods has brought about significant improvements, such as ultra-fast, portable integration, automation, and reduced sample sizes and reagent consumption, making them more suitable for clinical applications and ideal candidates by which to address the technical issues in liquid biopsy [69,70,71,72,73].

Here, the microfluidic systems for exosome isolation and biosensing approaches more suitable for integration in a single compact tool are reviewed and critically discussed.

### 3.1. Microfluidic Systems for Exosome Isolation

Conventional methods commonly used for exosome isolation from biological fluids and cell culture media include ultracentrifugation-based methods, size-based methods (size-exclusion chromatography and ultrafiltration), precipitation, and immunoaffinity capture [74,75,76,77]. Despite their extensive use for exosomes separation from different biological sources, even in combined approaches [78,79], they still show some limitations, such as the presence of many impurities, low recovery amounts, and modifications of the native properties of exosomes from their in vivo state; in addition, they are time-consuming and they need large amounts of samples and reagents.

Different types of new microfluidic devices have been designed, integrating laminar flow, secondary forces, external force fields, and unique geometries. Heterogeneous exosome populations can be separated using microfluidic systems solely based on their intrinsic properties (e.g., immunoaffinity-based exosome isolation, filtration and trapping separation, separation based on fluid properties), or on dynamic approaches, due to the application of an external field of forces (e.g., electroactive and acoustic separation, flow-based separation) (Figure 3). In both cases, the isolation process can be carried out through label and label-free approaches, with the latest being more promising for applications in highly sensitive and early-stage diagnosis.

#### 3.1.1. Microfluidic Systems Based on the Intrinsic Properties of Exosomes

##### Immunoaffinity-Based Exosome Isolation

Label-based isolation approaches exploit capture biosystems (e.g., antibodies and aptamers) able to chemically or physically bind specific lipid or protein molecules on the exosome out-layer membrane. It has been demonstrated that immunoaffinity-based platforms are able to isolate, with good purity, specific types of exosomes, including tumor-derived exosomes, from plasma, serum, and urine [80,81,82,83,84,85,86]. Patterned microstructures or nanostructures have been introduced in antibody-modified microfluidic devices to enhance the interaction between exosomes and chip interfaces [87]. The ExoChip platform implements multiple circular capture chambers interconnected by narrow channels, increasing the exosome retention time, and it is functionalized with antibodies against CD63, an antigen commonly overexpressed in exosomes. The use of the ExoChip to monitor exosome levels in pancreatic cancer patients was investigated [88]. A graphene oxide/polydopamine (GO/PDA) nano-interface was integrated into a microfluidic device, resulting in high-efficiency exosome immuno-capture and the suppression of non-specific exosome adsorption. The system was able to discriminate ovarian cancer patients from healthy controls by the quantitative detection of exosomes directly from 2 μL plasma without sample processing [89].

Recently, a clustered regularly interspaced short palindromic repeat (CRISPR)-associated proteins (CRISPR/Cas) system has been shown as a promising tool for the amplification and detection of specific biomolecules in extracellular vesicles. The isolation of exosomes through membrane protein recognition and signal amplification based on the CRISPR technique was applied to detect exosomes in clinical samples from patients with lung cancer [90].

These label-based techniques guarantee a high separation purity even from complex biological samples. However, they demonstrate low throughput and their application to multiple populations may be limited by the need-to-know molecular composition of the target exosomes. In addition, the label may interfere with exosome properties and biological activities, leading to a decrease in native information that has a fundamental role in diagnostic and therapeutic applications.

##### Label-Free Microfluidic Separation of Exosomes: Filtration and Trapping Separation; Fluid-Based Separation

The isolation of exosomes from biological fluids and purification from large particles and proteins can be achieved in microfluidic systems integrating nonporous membranes and pillar arrays in a label-free approach [91]. Recently, Zhenglin et al. presented a cascaded microfluidic circuit for the pulsatile filtration of particles directly from whole blood samples with high yield and purity within 45 min for fast cancer diagnosis based on liquid biopsy [92] (Figure 4I). The ExoTIC chip was specifically designed by Liu et al. to cause the biological samples (urine, blood, culture media) to pass through a nanoporous membrane, thus allowing an enrichment in particles with higher yield than ultracentrifugation [93].

Exosomes of a particular size can be selectively trapped, exploiting microstructures such as pillars and herringbone grooves into microfluidic systems [96]. The distance and morphology of these microstructures regulate the size separation of exosomes. A microfluidic device consisting of ciliated micropillars, forming a porous silicon nanowire-on-micropillar structure, was described, with potentialities for the isolation of exosome-like vesicles. Particles can be selectively trapped depending on the spaces among structures, while large components, such as cells, were filtered out [97] in about 10 min with approximately 60% recovery. Zinc oxide (ZnO) nanowire-anchored polydimethylsiloxane (PDMS) microchannels were proposed for the highly efficient separation of EVs from 1 mL of urine within 20 min [98]. Despite this relatively high efficiency, when volumes of samples higher than 30 μL were passed, the recovery significantly dropped due to the saturation of the pillar surfaces.

The size-based separation methods described here allow for the successful isolation of exosomes, with the limitation of an inability to distinguish other vesicles of different natures. In addition, complex biological matrices can lead to a saturation of microstructures with the consequent loss of separation capacity.

The physical constraints of specific exosomes in the streamline of a continuous flow in microchannels can be achieved by nano-pillar array design or the mechanical properties of the mean in a label-free approach. The design of the curved microfluidic channel can enhance the size-separation process due to the centrifugal effects [99]. Examples of microfluidic platforms based on this principle applied to exosome isolation are viscoelastic-flow sorting and deterministic lateral displacement (DLD) [100]. In the viscoelastic flow sorting, elastic lift forces act on particles of different sizes in a viscoelastic medium typically consisting of biocompatible and soluble polymers such as poly(vinylpyrrolidone), poly(oxyethylene), and polyacrylamide. A high-throughput and label-free sorting of exosomes from cancer cell line medium was demonstrated [94,100] (Figure 4II). In DLD, arrays of pillars are integrated into a flat microchannel to control the trajectory of particles in fluid flows along the length of the channel; thus, particles can be separated based on a cutoff diameter [101,102]. The size-based particle separation from both serum and urine samples was achieved with a technology that integrates 1024 nanoscale deterministic lateral displacement arrays on a single chip characterized by a high throughput [103].

Size-based separation with high resolution can be achieved by employing these microfluidic tools. However, these systems still require long processing times to isolate exosomes from biological fluids. In addition, due to the complex nanostructure, a loss of sample, particularly when intact biological matrices are analyzed, is involved. A loss of sample and a complex fabrication process are involved. Commonly, these systems require complicated fabrication processes that limit the rapid diffusion as a tool for the early-stage diagnosis of tumors.

#### 3.1.2. Microfluidic Systems Based on Dynamic Separation

All of the above-described systems based on a passive separation are, however, limited to highly specific isolation protocols and they cannot be applied to any biological fluids. Microfluidic systems where the separation is based on the application of an external field of forces (e.g., electroactive and acoustic separation, flow-based separation) may represent an interesting alternative, since they are based on a dynamic approach.

##### Electroactive and Acoustic Separation

Electrokinetically based microfluidic devices represent an attractive alternative for exosome separation from biological fluids without the need for pretreatment or dilution of the sample. For instance, electrophoresis was embedded in a microfluidic system to drive charged particles across a filtration membrane in order to sort them based on different sizes and charge-mass ratios, and to eliminate clogging and proteins [95] (Figure 4III). An electric field was applied across a dialysis membrane with a pore size of 30 nm, which was exploited to capture particles on the membrane surface and remove proteins, with high efficiency and a short analysis time, from plasma samples [104]. Dielectrophoresis (DEP) consists of the migration of electrically polarizable (dielectric) particles within a nonuniform electric field by applying alternating voltage on a microelectrode; thus, particles can be separated relative to their size and dielectric properties [105]. Microfluidic systems implementing the DEP process are widely used to selectively move nanoparticles from biological fluids [106,107,108]. Ibsen et al. proposed an electric microarray chip based on DEP technology to isolate the exosome of glioblastoma from undiluted blood samples and move cells and macromolecular proteins to a different region of dielectric electrophoresis [109].

Although nanoparticle isolation based on electric fields has shown promise, this approach still presents some disadvantages in terms of robust use for diagnostic applications. Mainly, the contact with the electrodes and the high operational voltage required may modify the native properties of biological samples. In addition, the applied electrical field influences the movement of all nanoparticles present in the biological fluid; thus, exosomes can be isolated together with impurities such as protein aggregates.

The use of acoustic wave technology can represent a label-free approach to sorting intact exosomes based on their native biomechanical properties, such as size density and membrane composition [110,111,112]. Electrodes on a piezoelectric substrate (e.g., quartz or lithium niobate) are used to generate the acoustic waves. Thus, biocompatible systems, avoiding the contact of the biological sample with the device, and leading to an easy integration with microfluidic components, can be obtained [113,114]. These systems may act as trapping approaches or be integrated into microfluidics for exosome isolation from biological fluids [111,115] (Figure 4IV). The proposed microfluidic devices require complicated designs utilizing multiple layers and small nanoscale structures, as well as surface modification to avoid nonspecific adsorption; thus, they appear suitable as routine tools for clinical applications.

##### Flow-Based Separation

Microfluidic systems based on the application of hydrodynamic forces represent a versatile option for the separation of bio-nanoparticles with high recovery.

Among flow-based sorting techniques for nanometer-sized analytes, field-flow fractionation (FFF) is widely used for the isolation of nanoparticles from biological matrices [116,117]. The most established FFF variant is flow field-flow fractionation (F4) where an external flow field is perpendicularly applied to the parabolic flow in an empty separative capillary channel (Figure 5A) [118]. In F4, retention is inversely proportional to the hydrodynamic diffusion coefficient of the analyte and, consequently, to its hydrodynamic size (Figure 5B). F4 can separate analytes based on their native form without the need for a label or manipulation with total maintenance of their native properties, including biological activity, surface, and phenotype properties. In addition, F4 can be applied in a wide dimensional range. These features make F4 a label-free and versatile technique suitable for the isolation of nanoparticles from complex biological matrices.

F4 has already been applied to the analysis of exosomes and vesicles from cell lines and biological samples [119,120,121,122,123,124]. F4 has been used to address the complexity of EVs from a cancer cell line through the study of their proteomic and genomic profiles, as well as biophysical properties [125]. This paper highlights the crucial role of the isolation process to elucidate which properties mainly influence metastatic patterning and the systemic effects of cancer. Recently, F4 has been used to demonstrate that a high number of vesicles can be isolated from human serum, promoting their biological studies [119] (Figure 5C).

From the perspective of microfluidic tool development, the micro-volume variant to F4, hollow-fiber flow field-flow fractionation (HF5), was demonstrated as being capable of achieving high performance and low dilution at the same time for biological particle analysis with interesting advantages: a reduced channel volume (in the order of 100 mL), low operation flowrates (as low as 400 mL/min), and potentially disposable use, which eliminates the risk of run-to-run sample carryover [126,127]. HF5 has already been proposed as a microfluidic tool integrated into analytical platforms for the fractionation of intact proteins from biological fluids and proteomic analysis from complex biological samples [127,128].

In HF5, separation is achieved in an empty capillary channel consisting of a tubular membrane with porous polymeric or ceramic walls (length = 17 cm, internal diameter = 0.8 mm), and by the combined action of a laminar flow of mobile phase and orthogonal flow (cross flow), which permeates the membrane. The cross flow represents the field needed for the separation and it can be tuned to optimize the separation process. In HF5 (as in F4), retention is inversely proportional to the hydrodynamic diffusion coefficient of the analyte and, consequently, to its hydrodynamic size. The separation mechanism and HF5 set-up are fully described elsewhere [129] and references therein (Figure 6I). HF5, coupled with online uncorrelated detection methods including Multi-Angle Light Scattering (MALS), absorbance, and luminescence spectrophotometry, provides highly resolved size distribution and differently sized subpopulations from different complex matrices [127,130,131,132,133]. 

We recently showed HF5 characterization in terms of size, abundance, and the DNA/protein content of subpopulations of membrane-derived vesicles from culture medium of murine myoblasts, and their purification into fractions for further biological characterization (Figure 6II) [134].

Due to the interesting features related to its miniaturized format and easy integration into microfluidic tools, from this perspective, we present the project for a miniaturized biosensing system based on HF5 for exosome isolation in native form and their cargo analysis in this paper (paragraph 4).

Flow-based separation tools represent one of the most promising approaches, since they can isolate exosomes in a native state by means of a versatile approach that can be modulated based on the type of biological sample employed. In addition, these systems can be easily integrated with specific biosensing modules to improve analytical data on exosomes. Some efforts have to be made in regard to their miniaturization through the implementation of miniaturized flow control and regulation systems.

### 3.2. Biosensing Approaches

Most of the exosome analysis devices are based on the molecular recognition of outer membrane molecules from the tetraspanin family of proteins (CD63, CD81, and CD9) using specific antibodies or other recognition elements [135]. Then, the exosome content, e.g., ‘cargo’, is usually detected after their lysis and conventional detection of the most relevant analytes, such as cytokines and other newly discovered biomarkers. Multiplex detection is usually required to achieve the required diagnostic power.

Exosome analysis, similar to exosome separation/purification, is widely open to improvements due to the limitations of the techniques commonly exploited for the task. Traditional approaches involve the use of flow cytometry, transmission electron microscope, NTA, DLS, and Western blot [136]. They need skilled operators and are sometimes laborious and time-consuming. On the other hand, biosensors are generally considered promising tools for sensitive, selective, reliable, cost-effective, and convenient analysis [137]. Biosensor development must consider the nature of its main components: the biorecognition element, (i.e., protein, aptamer, antibody, nucleic acid), which should provide selective marker targeting and the signal transducer/reporter. In this section, we discuss the advantages and limitations of the main transductor signal typologies in a miniaturized format highlighting whenever they have been applied to exosome detection (Table 1).

#### 3.2.1. Reagent-Based Systems

Colorimetric methods are one of the most user-friendly approaches in biosensing since the signal can be interpreted not only by a UV/Vis spectrophotometer but also by the naked eye (qualitatively). Overall, they represent an economical and practical approach with broad prospects in biomedical applications, though the methodology is characterized by a higher LOD [138]. Colorimetric sensors for exosome detection exploiting gold nanoparticles [139,140] or single-walled carbon nanotubes [141] as colorimetric indicators have been developed.

Chemiluminescence (CL)-based sensors are another large and heterogeneous family of biosensors that have been exploited in a variety of bioanalytical formats, including microtiter plate, microarrays, microfluidics, paper-based devices, such as lateral flow immunoassay (LFIA), and in vitro microscopy imaging [138]. CL is based on the production of photons triggered by a chemical reaction; when the reaction occurs within living organisms, the phenomenon is called bioluminescence (BL). Since BL and CL reactions start in the dark, photons can be measured with high efficiency, ensuring the absence of the background commonly encountered with photoluminescence, e.g., fluorescence measurements. Additionally, they do not require complex instrumentations, making them ideal detection principles for point-of-care (POC) devices [142]. One of the main issues is that they rely on additional reagents, which are also not stable and require dedicated shipping and storage conditions [143]. When CL is induced by the application of an electric potential, the phenomenon is called electrogenerated chemiluminescence (ECL). The process is controlled by the applied potential, allowing precise modulation over the time and position of the emission, allowing for photo collection efficiency and reaction specificity optimization [144]. Since the emitted light is generated by a different physical principle, ECL allows for the development of sensors characterized by a very high signal-to-noise ratio, though they still require external reagents (it is not without reagents) and its miniaturization is limited by the electrode miniaturization capability [138]. Overall, ECL has been the CL phenomenon most exploited to develop sensors able to detect and quantify exosomes [145].

#### 3.2.2. Reagent-Less-Based Systems

Electrochemical (EC) biosensors are a wide family of biosensors that can convert the recognition of a biomolecule into electrical signals (current, potential, or impedance). They exploit electrodes functionalized with tags designed to generate the electric signal while interacting with the analytes [146]. Overall, these systems are characterized by high sensitivity, rapid response, good multiplexing capability [147], and easy miniaturization thanks to integrated circuit technology. These characteristics allowed for their exploitation in exosome analysis, which has been widely documented [148,149]. The main problems of these devices are instead represented by the difficulties in electrode surface actualization and the common presence of the matrix effect [145].

Surface plasmon resonance (SPR) is a label-free, reagent-less, real-time analysis technique that can detect molecular interactions on the surface of a gold layer by monitoring the changes in its refractive index [150]. Several biosensor platforms integrating SPR have been developed with excellent performances in terms of low sample consumption, multiplexing analysis, specificity, and sensitivity for disease-specific exosome analysis [145]. Some issues, such as such as the nonspecific adsorption of sample components on the sensor, however, need to be overcome to make the SPR technology fully suitable for clinical application. Moreover, at present, most of these systems are still in the proof-of-concept state or have not been properly validated [151].

Another innovative approach for biosensor technology is based on surface-enhanced Raman scattering (SERS). As the name suggests, this phenomenon is charactered by an enhancement of the Raman scattering signals of molecules when placed in the vicinity (<10 nm) of some nano-materials, including plasmonic noble nanostructures, 2D nanomaterials, and semiconductors [152]. SERS-based sensors have received a lot of attention in biomarker analysis (such as exosomes) due to their sensitivity and a low signal-to-noise ratio background, as well as their non-invasive nature and requirement for very low amounts of sample [153]. The SERS sensors for EV analysis can be classified into two categories: label-based and label-free [154]. The use of SERS tags allows for the ultra-sensitive detection of analytes and relatively simple manufacturing, but it does not provide any structural information. On the other hand, label-free SERS, through the Raman fingerprint signal of the analytes (caused by their proximity to the SERS platform), can provide structural information about the target sample, but it suffers from difficult data processing. At present, although both systems are characterized by high reproducibility and low detection limits, they necessitate complex instrumentation to perform the measurements and analyze the data, making them not ideal for point-of-care applications [155].

Fluorescent molecules have been one of the most popular reporters/transducers due to their high sensitivity, rapid response, adaptability, and the fact that they do not require additional reagents besides the analyte to perform the assay [148]. A problem hindering their application (i.e., point-of-care applications) is the complex instrumentation required to perform the measurements, which is comprised of an excitation light source and critical detector geometry [156]. Additionally, fluorescence-based sensors may show low sensitivity, accuracy, and a high background in biological samples due to the natural fluorescence of different compounds. A solution to the problem is represented by the time-resolved fluorescence technique (TRF) [157]. TRF applies the temporal domain to differentiate targets labeled with long-lifetime fluorescence from short-lifetime autofluorescence [158]. At present, both fluorescence sensor types have been developed for exosome analysis [74,159,160] and are commercially available.

Another CL mechanism exploited in sensors is Thermochemiluminescence (TCL). TCL is based on the emission of photons as a consequence of the thermolysis of a suitable molecule used as a label [161]. The main advantage of this approach is that no additional reagent is required to trigger the light emission from the TCL label. The poor performances of the first pioneering devices (exploiting 1,2-dioxetane derivatives as labels) led to an early abandonment of the techniques, which has recently been resumed to overcome its limitations. New TCL molecules with enhanced photophysical and fluorescence properties have been recently synthesized by us and used as a probe to label biomolecules included in silica nanoparticles (SiNPs) to increase their stability and analytical performances [162].

More recently, the TCL molecules have been included in semiconductive polymer dots (PDot) nanoparticles, providing improved detectability with remarkable stability over time and minimum leaching of the thermos-responsive species [163]. Their first application as biosensors were related to the detection of valproic acid [164].

The TCL system presents a LOD comparable to those obtained using enhanced HRP-catalyzed CL detection [165].

## 4. Towards a New FFF-Based Multiplex Biosensor

Despite the fast development of new analytical formats for biosensors based on microfluidic systems and reagent-less detection, as described in the previous paragraphs, the development of a robust, portable, and easy-to-handle biosensor able to directly detect specific exosomes and cargo from a biological fluid is yet to be achieved. The crucial step limiting the development of these biosensors is their poor applicability to exosome analysis from intact complex biological samples. Improved analytical tools should be able to perform all the steps of the analysis, including sample pre-treatment, reagent delivery, mixing, separation, and detection, and be automated and integrated into a miniaturized chip format.

As a step forward from the state of the art, we first explored the use of miniaturized HF5 for the separation and isolation of exosome fractions according to their size and morphology from the undiluted serum of colon cancer patients. Then, following our previous experience in many different analytical formats of ultrasensitive biosensors based on luminescence and the development of a new generation of bioanalytical formats using HF5, we designed a new multiplex microfluidic biosensor where an online HF5 preanalytical step was able to directly enrich exosomes from serum integrated with a biosensing module for intact exosome quantification and multiplexed cargo analysis. For detection approaches, we chose reagent-less luminescence-based technologies TRF and TCL based on acridine-doped SiNPs [166]. As a proof of principle, the system was applied to serum samples from patients affected by colorectal cancer (CRC) and healthy donors (HD). Among exosome markers, we employed the membrane-specific CD9 protein, an endosome-specific tetraspanin of the exosome membrane, for intact exosome detection, and interleukin 6 (IL6) as a cargo biomarker.

### 4.1. Exosomes Isolation and Cargo Analysis

#### 4.1.1. Study Design and Serum Sample Collection

The study was performed with approval from the Local Ethics Committee of the Emilia Romagna region (601/2018/Sper/AOUBo) and followed the Declaration of Helsinki and its later amendments. This was an exploratory, controlled, single-center human study, which is still ongoing at the Sant’Orsola Malpighi Hospital in Bologna, which identified three groups of subjects: 10 subjects with hyperplastic polyps, 20 with low- and high-grade adenomatous polyps, and 10 subjects with colorectal cancer. Enrolled subjects, which provided written informed consent, were patients that routinely undergo colonoscopy exams and are treated according to clinical practice. Whole blood samples were collected from each subject during the normal course of care in accordance with pre-processing guidelines for EV-based biomarker analysis [167].

#### 4.1.2. HF5 Instrumental Setup and Exosomes Isolation Performances

Before the integration into the biosensor microfluidic system, the separative performances of HF5 in conditions compatible with the biosensor requirements (i.e., sensitivity, sample amount) were verified using a complete HF5 multi-detection platform [168]. The separation method was adapted from [169] in order to reduce separation in the molecular (30–200 k Da) range typical for proteins and to optimize the separation window of species with a diameter higher than 20 nm. Then, 30 μL of serum was injected in HF5 after 1:2 dilution in physiological PBS. UV detection was set at 280 nm, typical of protein absorption maxima. Multiangle light scattering (MALS) allowed us to calculate the molar mass of eluted species and confirm the dimension of isolated particles. Three repeated injections of each sample were performed, and fractions of size-based separated eluted particles were collected, lyophilized, and stored for further determination of an exosomal nature.

This preliminary study was performed using three samples from HD and CRC individuals, respectively.

The HF5 separation profiles are reported in Figure 7A,B. Both sample types showed identical separation profiles, composed of two main bands eluted at 5 and 11.5 min. The first peak contained mainly albumin and immunoglobulins, as confirmed by injections of HSA-IgG mixes (dashed lines) showing a similar retention time. The mass calculation averaged 140 k Da, coherently with the attribution. The second band corresponded to a molar mass distribution starting from 800 k Da and reaching the tens of million Da, tailing until minute 20.

The HF5 method was also calibrated through FFF theory to visualize the dimensional distribution of fractionated samples. From the correspondence between the retention time and the hydrodynamic radius (graphed in Figure 7C, top), the second peak was determined to contain species with a diameter ranging from 15 to 120 nm: by correlating this information, it was confirmed that the second band contained IgA, lipoproteins, and, at higher times, vesicles and vesicle-like particles [169].

To determine the collection window of exosomes to be subsequently delivered to the integrated biosensor, the fractogram was split into three sections, corresponding to: i. smaller serum proteins; ii. particles between 15 and 50 nm; and iii. particles between 50 and 120 nm; the three fractions were then collected. The offline quantification of exosomes marker CD9 analysis was performed on lyophilized fractions using a microtiter immunoassay with specific Eu-Ab and TRF measurements. All samples were resuspended in water at the same initial volume of serum injected, to have an immediate correspondence between concentrations calculated for fractions and unfractionated serum. CD9 quantification (Figure 7D) showed that fraction one did not significantly contain CD9, as expected, and removal of this portion from serum fractionation is indeed beneficial to quantification. Instead, fraction 2 contained a significant amount of CD9-positive particles, and more so for CRC samples. The expression of CD9 is higher in exosomes derived from malignant samples compared to healthy samples, thus it could represent an outer membrane marker for exosome quantification in cancer samples [170]. Last, fraction 3 contained CD9-positive fractions, but with an inverted trend between the HD and CRC samples. Interestingly, the results on unfractionated serum showed a lower concentration of CD9 and lower reproducibility, indicating that interfering proteins could play a role in marker quantification.

The results proved that it was possible to isolate, through HF5, a fraction (fraction 2) where CD9 positive particles were enriched, while removing interferences. Surprisingly, this fraction corresponded to particles with a lower radius than expected, indicating that a native approach to sort whole particles can also help to achieve a better understanding of serum-derived exosomes and their cargo. The chance of selectively convoying particle-rich serum fractions to the microfluidic sensor would be invaluable in both optimizing exosome capture, increasing sensitivity, and reducing sensor degradation from interfering agents.

#### 4.1.3. HF5-Based Microfluidic Tool: Simultaneous Exosomes CD9 Membrane Protein and IL6 Quantification

Once verified, the HF5 isolation performances, as a proof of principle for the detection of exosomes through their specific CD9 membrane protein, followed by the specific IL6 quantification in their cargo, was shown through the designed microfluidic system with the online HF5 isolation step.

A schematic representation of the proposed device is reported in Figure 8. Briefly, the HF5 is online, connected using a capillary tube and a valve to split the flow. The fractogram interval enriched in exosomes is directed to a detection chamber where magnetic nanoparticles coated with anti-CD9 antibody (NP-Ab-CD9) and secondary Eu-Ab are added and left to incubate for about 15 min. Hence, under a magnetic field to retain the immunocomplex NP-Ab-CD9-exosome-Eu-Ab, the chamber is washed with physiological PBS and the TRF of the Ab-Eu bond is measured with a miniaturized portable TRF instrument. After exosome detection, a urea lysis buffer solution is delivered through the microfluidic system to destroy the exosome membranes and release cargo [171]. The solution is then transferred to the cargo multidetection chamber constituted of an array of up to 9 microwells where specific antibodies for different cargo biomarkers are immobilized. Although the microfluidic biosensor may allow for the multiplex detection of different cargo biomarkers, as a proof of principle, we explored its use for one analyte. We chose IL6, since exosomes in the tumor microenvironment can activate interleukin-related mechanisms in cancer, including CRC. After a short time for the immunoreaction, the microfluidic system transfers a secondary Ab labeled with the TCL-doped SiNPs [162]. The wells are promptly heated with a mini heater pad (Watlow series Ultramic) at 120 °C with an ITO pad placed under the well and the emitted light is measured with a CMOS (smartphone or similar device).

Figure 9 shows the very promising results of six HD and CRC samples, obtained with the HF5-based microfluid devices. Representative TRF and TCL imaging signals for the control and CRC patients (Figure 9A) and CD9 and IL6 quantification are reported (Figure 9B). Higher values for CD9 and IL6 were detected in CRC samples; indeed, increased amounts of exosomes and higher levels of IL6 in patient samples may represent specific biomarkers for the early diagnosis of colon cancer, as presented in paragraph 2.

These preliminary and limited results suggest the potential application of the proposed HF5-based biosensor in liquid biopsy. More than one cargo biomarker can be detected, since multidetection can be achieved through the immobilization of an array of capture antibodies in the immunoassay chamber. Studies have to be conducted in order to avoid cross-reaction and provide selective spatial detection. In addition, the integration of the miniaturized HF5 to the microfluidic biosensor can be improved through the use of a microfluidic pump able to accurately manipulate pulse-free fluid in minute volume below the microliter range, including a pressure and flow controller in the same device [172]. From the perspective of a compact miniaturized chip format for the biosensor, the same detector can be used for TRF exosome and TCL cargo quantification. A smartphone could be equipped with a simple time-resolved fluorimeter, a pulsed laser source, and delayed luminescence acquisition to only measure the emitted light from the lanthanide chelate probe when the endogenous fluorescence is extinct [173].

## 5. Conclusions

Cancer-derived exosomes serve as biomarkers for the early and non-invasive detection of cancer, as they carry the cargo reflective of genetic or signaling alterations in cancer cells of origin. Highly sensitive and specific diagnoses can be obtained by means of the simultaneous detection of intact exosomes and their specific cargo, after their purification from the biological samples. Currently, there are no gold standard methods for the isolation of exosomes. To improve the use of exosomes in clinically meaningful tests, isolation procedures should guarantee exosome purity and the preservation of the exosome structural integrity and biological activity to allow their proper characterization and profiling. Sample contaminants have to be eliminated and sheer stress, which may activate uncontrolled biological pathways, should be reduced. Microfluidic and flow-based systems in a label-free approach may address these issues. Among them, HF5 has emerged as a powerful miniaturized isolation technique able to separate size-based exosomes from the biological matrix in its native form.

In the future, the development of versatile, robust, and simple microfluidic-based devices will be achieved through the integration of microfluidic pump and valve systems, innovative separation, and biosensing tools, including a specific biomolecule detection step based on a reagent-less approach and disposable CMOS or silicon photodiodes systems. In this configuration, the microfluidic biosensors can be easily automated for the simultaneous detection of intact exosomes and their cargo from a multianalyte perspective. In addition, the integration of machine learning (ML) and artificial intelligence (AI) may also improve the design and development of standardized and multiplexed analytical tools. ML and AI can be used as control systems by which to standardize the analytical procedure and to manage obtained data, making the microfluidic biosensor a robust tool for improved diagnosis in cancer based on liquid biopsy with interesting perspectives of personalized medicine.

## Figures and Tables

**Figure 1 sensors-23-09432-f001:**
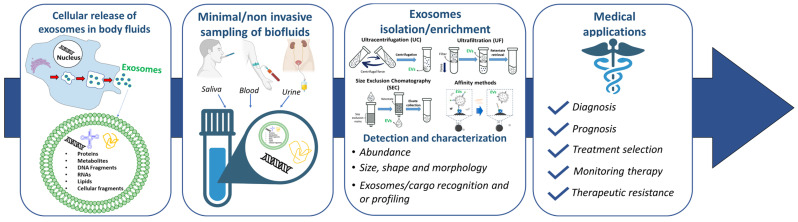
Principle and workflow behind the liquid biopsy of exosomes.

**Figure 2 sensors-23-09432-f002:**
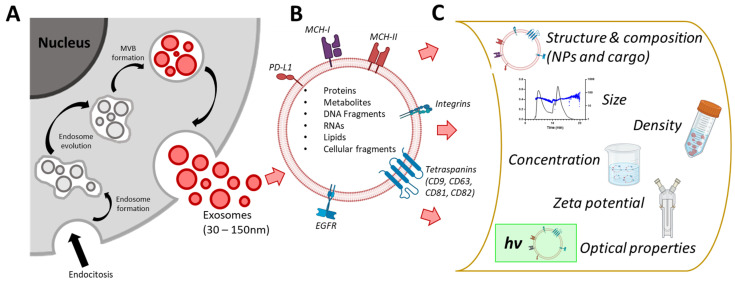
Overview on exosomes. (**A**): biogenesis. (**B**): structure and cargo composition. (**C**): properties that should be evaluated for a complete characterization. Adapted from [4].

**Figure 3 sensors-23-09432-f003:**
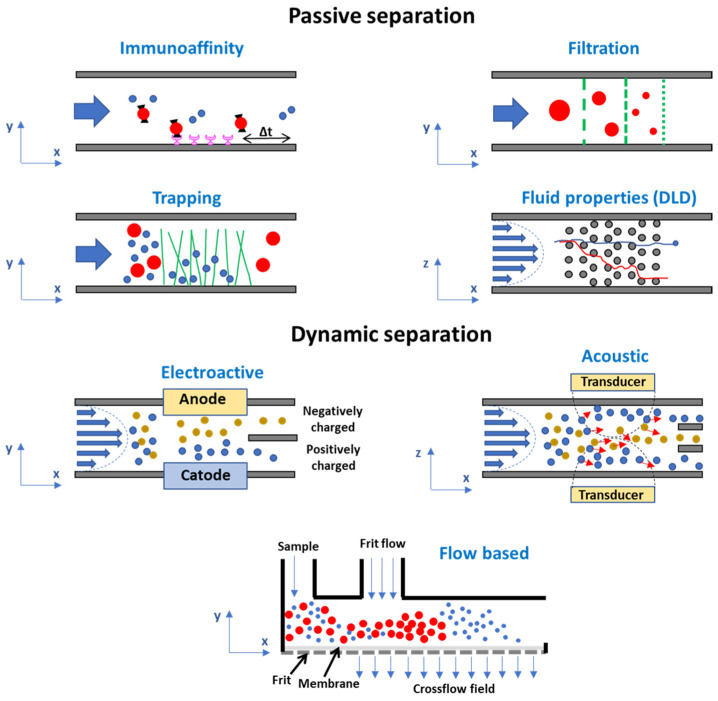
Schematization of the main microfluidic technologies for exosome isolation.

**Figure 4 sensors-23-09432-f004:**
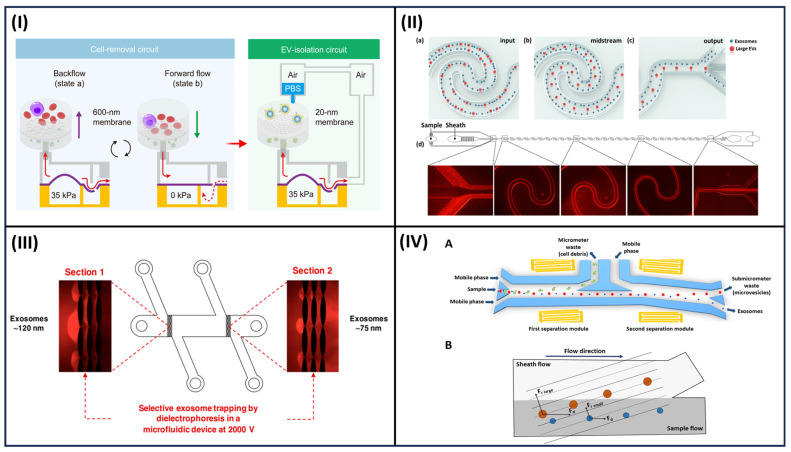
Examples of the label-free microfluidic separation of exosomes. (**I**) Cascaded microfluidic circuits for vesicle isolation from whole blood. Adapted from [92]. (**II**) Size-based elastoinertial exosome sorting device. The microfluidic periodically reversed Dean secondary flow generated by repeated curvilinear channel structures for particle focusing. The label-free sorting of exosomes with purity higher than 92% and recovery higher than 81% can be achieved. Adapted with permission from [94] “Copyright (2019) American Chemical Society.” (**III**) Microdevice for the direct current–insulator-based dielectrophoretic (DC-iDEP) approach to simultaneously capture and separate exosomes by size. The microdevice consists of a channel with two electrically insulating post sections generating different nonuniform spatial distributions of the electric field. By applying an electric potential difference of 2000 V across the length of the main channel, the dielectrophoretic size-based separation of exosomes was observed in the device. Adapted with permission from [95] “Copyright [2019] American Chemical Society.” (**IV**) Acoustofluidic device for salivary exosome separation (**A**). The device has two modules using 20-MHz and 40-MHz surface acoustic waves (SAWs). Due to the acoustic radiation force induced by the SAW field and drag fluid, large particles are constrained into a sheath flow, whereas smaller particles remain in the primary sample flow (**B**).

**Figure 5 sensors-23-09432-f005:**
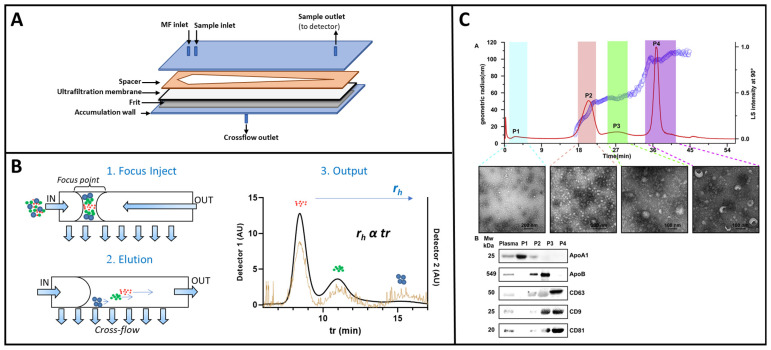
(**A**) Schematization of an AF4 channel: an ultrafiltration membrane is placed on a spacer with a typical thickness of 250–800 µm. A porous frit of ceramic or metal material is placed under the filter membrane (accumulation wall). Polycarbonate walls are used to assemble the layers. (**B**) Main steps of an AF4 separative protocol and final output: 1. During the focus-injection step, analytes are equilibrated in a narrow band at the beginning of the channel. 2. In the elution step, the flow (IN) is split into two components, a longitudinal laminar flow (with a parabolic profile) named the detector flow and a perpendicular flow named the crossflow, the driving separation is based on the hydrodynamic radius of the analytes. 3. The output of the system, called the fractogram, is represented by the signals of the separated analytes collected over time by the detectors coupled to the channel. (**C**) Representative AF4 fractionation profile of human plasma-derived extracellular vesicles. Hydrodynamic radius distribution from the MALS, TEM imaging of particles, Western blotting analysis confirm vesicle fractionation. Adapted from [119] with permission.

**Figure 6 sensors-23-09432-f006:**
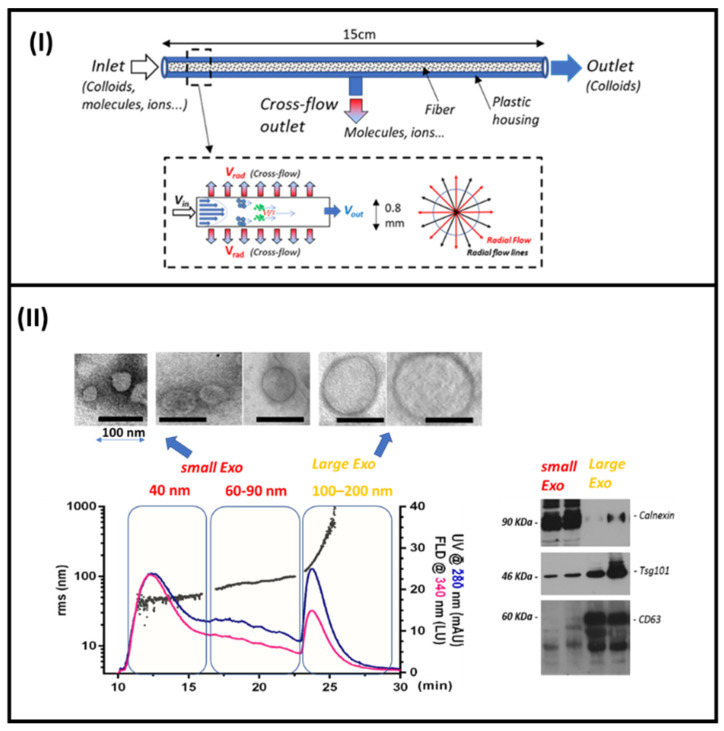
(**I**) HF5 microcolumn and separation mechanism. (**II**) HF5 separation of different exosomes from C2C12 cell culture medium. Western blotting of isolated fractions of small and large exosomes. Adapted from [134].

**Figure 7 sensors-23-09432-f007:**
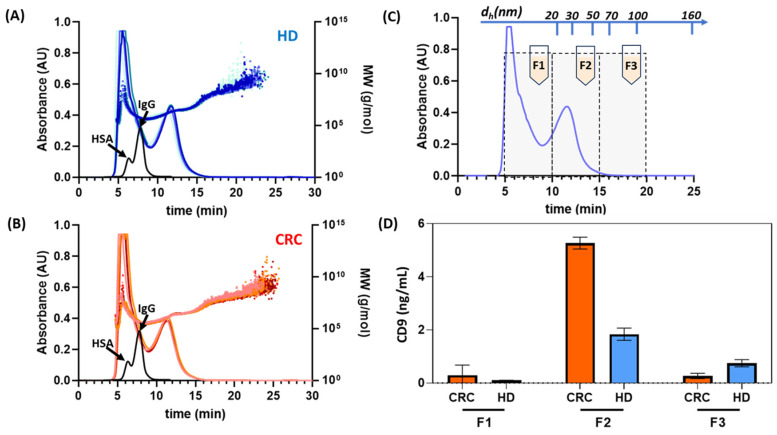
HF5 analysis of healthy and CRC serum samples. (**A**) UV fractograms (blue/azure lines) resulting from the separation of healthy (HD) serum overlapped with the corresponding calculated mass distribution (dotted blue lines). (**B**) UV fractograms (red/orange lines) resulting from the separation of CRC serum overlapped with the corresponding calculated mass distribution (dotted red/orange lines). The black line represents the fractogram resulting from the separation of an HAS-IgG mix with the same method. (**C**) Method size calibration and fraction intervals overlapped over a representative serum UV fractogram. (**D**) CD9 quantification for fractions F1, F2, and F3 (section C) of HD and CRC samples.

**Figure 8 sensors-23-09432-f008:**
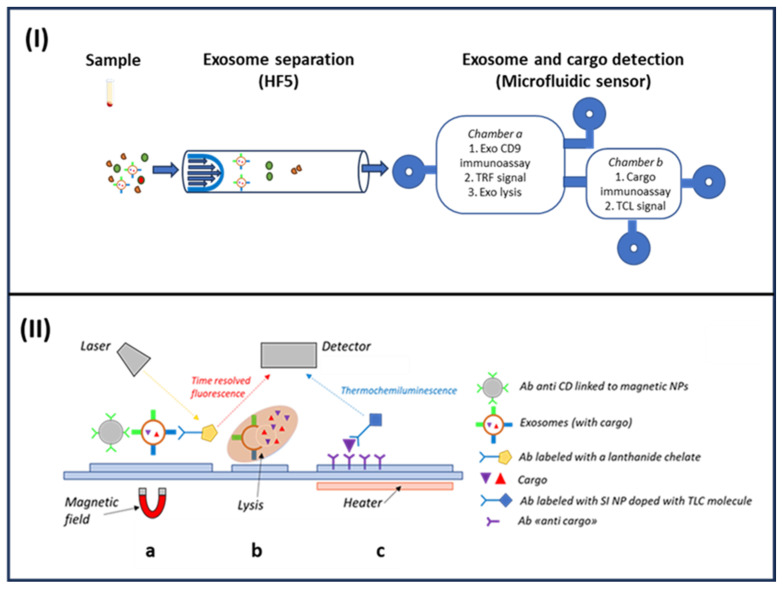
(**I**) Schematic design of the microfluidic biosensor. HF5 fractionation of serum sample delivers intact exosomes to the microfluidic sensor. Intact exosomes are quantified, and their cargo is detected: in *Chamber **a***, antibodies are added for the intact exosome’s detection through immunoassay and the TRF signal is measured; then, reagents are added for exosome lysis. The lysed sample is moved to *Chamber **b*** where antibodies are added for cargo detection through immunoassay, and the TCL signal is measured. (**II**) Materials on the microfluidic biosensors: (**a**) magnetic nanoparticles coated with anti-CD9 antibody (MgNP-Ab-CD9), and secondary Eu-Ab are added and left to incubate. The immunocomplex MgNP-Ab-CD9-exosome-Eu-Ab is retained under a magnetic field and the TRF signal of the Ab-Eu bond is measured. (**b**) A urea lysis buffer solution is then delivered and (**c**) the solution is transferred to the cargo multidetection chamber. Antibodies labeled with TCL-doped SiNPs are added and cargo content is detected through an immunoassay and TCL signal measurement.

**Figure 9 sensors-23-09432-f009:**
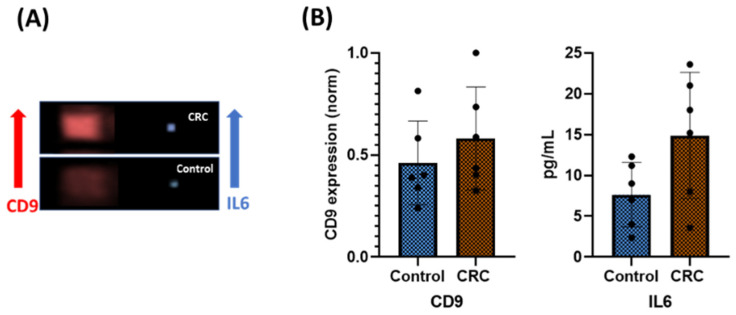
Intact exosome identification (CD9) and cargo quantification (IL6) in control and CRC patient serum. (**A**) Representative TRF (CD9) and TCL (IL6) imaging signals for control and CRC samples; (**B**) quantification of CD9 expression and IL6 in control and CRC samples. The graph shows the mean ± standard error of the mean (SEM) of the results for control (HD) and patient (CRC) samples (*n* = 6 for each condition). *p*-value < 0.05.

**Table 1 sensors-23-09432-t001:** Biosensing approaches for microfluidic-based exosome analysis.

Biosensing Approach	Advantages	Limitations
Optical	Rapid response, easy to use, inexpensive, qualitative naked-eye detection, POC capability.	Low sensitivity, limited multiplexing capability.
EC	High sensitivity, rapid response, Inexpensive, multiplexing capability, reagent-less	Challenging surface functionalization, matrix effect, reproducibility problems
Fluorescence	High sensibility, rapid response, multiplex capability, reagent-less	Complex instrumentation required, high background for complex bio samples
Fluorescence (Time resolved)	High sensibility, rapid response, high selectivity, multiplex capability, reagent-less	Complex instrumentation required.
CL/BL	High sensibility, rapid response, POC capability	Reagent-dependent,
ECL	Higher s/n and specificity compared to CL	Challenging miniaturization, reagent-dependent
TCL	Reagent-less	Developmental stage technology
SPR	High sensitivity, real-time detection, label-free system	Non-specific absorption, proof of concept state, complex equipment required
SERS (Label-aided)	Superior sensitivity, multiplexing capability, simple manufacturing	Costly equipment, difficult data analysis
SERS (Label-free)	Provide additional structural information of the analyte	Very complex data analysis

## Data Availability

Data are contained within the article.

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
