# Peer review of "Emerging Microfluidic Tools for Simultaneous Exosomes and Cargo Biosensing in Liquid Biopsy: New Integrated Miniaturized FFF-Assisted Approach for Colon Cancer Diagnosis"

_sensors, 2023, doi:10.3390/s23239432_

Round 1
Reviewer 1 Report
Comments and Suggestions for Authors
- There are some typos errors and grammatical issues that needs further consideration.
- Several sentences through the manuscript lack suitable citations.
- Please use conventional and routine terms for exosome biogenesis pathway. For example use multivesicular bodies instead of multivesicular endosomes
- The source of Figure 2 should be indicated whether designed by authors or used as permitted materials.
- The difference in exosome biogenesis between the healthy and cancer cells should be briefly compared as possible.
- The applicability of exosome evaluation in faces of CRC patients should be also discussed inside the manuscript as possible.
- The prominent disadvantages and advantages of microfluidics in terms of exosome isolation should be highlighted.
- The molecular signature of release exosomes and parent CRC cells can be explained more in this manuscript.
Comments on the Quality of English Language
- There are some typos errors and grammatical issues that needs further consideration.
Author Response
The authors want to thank Reviewer 1 for having revised our original manuscript. All comments have been accepted and revision performed accordingly.
- There are some typos errors and grammatical issues that needs further consideration.
We thank the referee. The text has been checked and typos errors have been corrected, and grammatical issue fixed.
- Several sentences through the manuscript lack suitable citations.
We thank the referee. References have been added in order to make the bibliographical references list more consistent and useful for covering all topics.
- Please use conventional and routine terms for exosome biogenesis pathway. For example use multivesicular bodies instead of multivesicular endosomes
We thank the referee. Conventional terms were included in this paragraph. In addition, we have modified the paragraph by adding details on exosomes biogenesis and release, and differences between healthy and cancer cells.
- The source of Figure 2 should be indicated whether designed by authors or used as permitted materials.
Figure 2 was designed by us starting from already published materials. Reference to the article was added to figure caption.
- The difference in exosome biogenesis between the healthy and cancer cells should be briefly compared as possible.
We thank the referee for the comment. A brief paragraph describing the exosomes biogenesis was added.
The hypothesis of biogenesis processes mechanisms for exosomes release in cancer cells were added and discussed. This also help to highlight the high heterogeneity of exosomes function and composition.
- The applicability of exosome evaluation in faces of CRC patients should be also discussed inside the manuscript as possible.
As far as we know the isolation of exosomes from human fecal samples remains unexplored due to their low levels in stool and the matrix complexity. However, the development of new microfluidic tools, especially through HF5, could be explored in the future as a promising approach to isolate fecal exosomes from human samples which have never been reported in literature so far for CRC studies. In this review, we focalized in microfluidics tools already applied to biological fluids for applications as liquid biopsy.
- The prominent disadvantages and advantages of microfluidics in terms of exosome isolation should be highlighted.
We thank the referee. We added some sentences at the end of each paragraph to highlight the advantages and disadvantages of the specific microfluidics tool described.
- The molecular signature of release exosomes and parent CRC cells can be explained more in this manuscript.
We thank the referee for this comment; this review was focused on exosomes released from colorectal cancer
cells and therefore no attention was given to the molecular signature of parent CRC cells. However, as requested by the Reviewer, we also wrote a brief overview of the molecular signature of CRC cells while emphasizing the most predictive biomarkers contained in the exosomes. Understanding the molecular signature of exosomes released by colorectal cancer cells is crucial for elucidating their involvement in cancer progression and potential diagnostic or therapeutic applications.

Reviewer 2 Report
Comments and Suggestions for Authors
The review is mainly focused on the exosome separation and detection methods using microfluidic techniques. For the separation part, could the authors illustrate more the necessity to separate exosomes first (with data proof would be the best); for the detection part, a recent popular technique using CRISPR was not included. For the experimental section, there are the following comments and questions to be answered or addressed before publishing:
Line 715: Could it be illustrated the difficulties and possibilities to avoid off-line analysis, but instead, as the workflow in Fig.7 demonstrated to directly couple the microfluidic device with HF5 for automated processes?
How necessary to separate EVs first to quantify the markers such as CD-9 and IL6? Closely viewing Fig.8D and Fig.9, I couldn't see what would be the difference if detection was performed in pooled EVs without separation.
What is the wash and lysis condition? Would it inhibit the post-RNA analysis?
Author Response
Rev 2
The authors want to thank Reviewer 2 for having revised our original manuscript. We address the comments below and based on these comments we changed and/or clarified several points of the manuscript.
The review is mainly focused on the exosome separation and detection methods using microfluidic techniques. For the separation part, could the authors illustrate more the necessity to separate exosomes first (with data proof would be the best); for the detection part, a recent popular technique using CRISPR was not included. For the experimental section, there are the following comments and questions to be answered or addressed before publishing:
We thank referee. We have expanded the description of biogenesis and characterization of exosomes (2.1. Exosome biogenesis and biophysics), even in tumor cells, giving greater emphasis to the heterogeneity of exosomes and the need for the separation both from biomolecules of different nature present in biological samples and among the different subtypes of exosomes, to obtain a precise characterization for an improved diagnostic approach. We also add some comments at the begging of 3. Integrated microfluidic system for exosomes analysis paragraph to highlight the need for exosomes separation before their characterization, and to summarize the main issues related to separation from impurities or biomolecules of different origin.
We add a comment on CRISPR technique to be used as improved sensing approach for label-based detection of exosomes. (Line 312-315, paragraph 3.1.1)
Line 715: Could it be illustrated the difficulties and possibilities to avoid off-line analysis, but instead, as the workflow in Fig.7 demonstrated to directly couple the microfluidic device with HF5 for automated processes?
We completely re-organized section 4 on HF5-based biosensor description. In this critical review we wanted to add the first preliminary results obtained with the on-line HF5 microfluidic separation system connected to the microfluidic module of the biosensor whose scheme is shown in figure 8 of the revised version of the manuscript. Before reporting these results, we wanted to demonstrate the suitability of HF5 for the direct isolation of intact exosomes from CRC serum samples; and this was done by injecting serum samples in HF5 device, collecting fractions and analyzing separated particles using CD9 as model marker for exosomes. Finally, we described the application of the on-line HF5 system for CD9 identification of isolated exosomes and IL6 cargo quantification and we demonstrated to be able to detect differences between healthy and cancer samples. We hope that in the revised version of the manuscript the message can be better conveyed to readers.
How necessary to separate EVs first to quantify the markers such as CD-9 and IL6? Closely viewing Fig.8D and Fig.9, I couldn't see what would be the difference if detection was performed in pooled EVs without separation.
We apologize with the reviewer for some missing considerations. First, we chose CD9 and IL6 as examples markers for intact exosomes and cargo content. Our final goal was to show as a proof of principle the use of an HF5-based microfluidic biosensor for exosomes analysis from complex biological sample. The HF5-separation step allows for the elimination of contaminants present in serum which could interfere with the biosensing module through the saturation of the microstructures, or the interference with the signal detection with improved perspectives for multiplexing applications. In addition, HF5 is able to size separate the different particles present in serum sample. Indeed, in order to develop a specific diagnosis even with perspectives in personalized medicine, it’s important to be able to detect the membrane markers and cargo contents of specific exosomes and avoid possible contaminations from other types of extracellular vesicles. For these reasons, it’s important to fractionate sample, to isolate only exosomes particles. In a future development, the fractionation will be able also to allow a correlation between specific cargo and exosomes subtypes making liquid biopsy a specific and precise powerful diagnostic tool. In the revised version of section 4 we tried to improve the discussion of these points. We thank reviewer for the comment, we will expand this point in the future development of the proposed biosensor with also some additional focalized experimental work.
What is the wash and lysis condition? Would it inhibit the post-RNA analysis?
A physiological PBS wash solution was used to isolate the immunocomplex held by magnetic field before signal detection. Urea lysis buffer (8 M urea buffer in 50 mM ammonium bicarbonate pH 8.5) was used as lysis buffer (see cited reference). We added some details on the text. These conditions with the use also of inhibitors are employed in reported work on proteomic analysis and molecular pathways identification in exosomes from cancer samples. However, we will consider this point for the future validation of proposed biosensor.

Reviewer 3 Report
Comments and Suggestions for Authors
The review article "Emerging Microfluidic Tools for Simultaneous Exosomes and Cargo Biosensing in Liquid Biopsy: New Integrated Miniatur-3 ized FFF-Assisted Approach for Colon Cancer Diagnosis" summarizes the recent advances in microfluidic systems for exosome isolation and detection, in addition to preliminary data on the hollow-fiber flow field-flow fractionation (HF5) system.
I believe that in its present form, the article is unsuitable for publication, mainly because of the excessive focus on presenting data related to the HF5 system, which detracts from the primary objective of a review article. Furthermore, when the results are presented, they are not structured and discussed properly.
Below are my specific comments:
1. A review article should discuss the current state and potential future advancements of a well-defined field or topic. Notably, a significant portion of section 4 is dedicated to presenting the working principles and results of a new biosensor.
2. The conclusion reads more like a research article, focusing on the HF5 sensor, rather than summarizing insights from the field.
3. If the data is presented in an article, it should be laid out in a proper manner with methodology, materials, results, and discussion sections. Placing everything in the introduction is not appropriate.
4. Exosome isolation requires more rigorous validation. For example, In addition to the CD9 analysis, it is critical to perform protein analysis on the isolated exosomes to confirm the presence of other exosomal markers (such as TSG101, CD63 and CD81) and the absence of non-exosome markers (calnexin). Furthermore, the characterization of the isolated exosomes' size and morphology is noticeably absent.
5. Figure 7-9 require revision. The clarity and image resolution of Figure 7 should be improved. Many elements within the figure are undersized and inadequately labeled, making it hard to understand. The figure legends in Figures 8 and 9 lack sufficient details.
Author Response
Rev 3
The authors want to thank Reviewer 3 for having revised our manuscript. We address the comments below and based on these comments we changed and/or clarified several points of the manuscript. We hope that the revision work performed following these comments have enhanced the quality of the revised paper.
The review article "Emerging Microfluidic Tools for Simultaneous Exosomes and Cargo Biosensing in Liquid Biopsy: New Integrated Miniatur-3 ized FFF-Assisted Approach for Colon Cancer Diagnosis" summarizes the recent advances in microfluidic systems for exosome isolation and detection, in addition to preliminary data on the hollow-fiber flow field-flow fractionation (HF5) system.
I believe that in its present form, the article is unsuitable for publication, mainly because of the excessive focus on presenting data related to the HF5 system, which detracts from the primary objective of a review article. Furthermore, when the results are presented, they are not structured and discussed properly.
We thank referee for the comments. We thank referee for the comment for letting us realize that we put too much focus in the first preliminary results on the new HF5-based microfluidic tool. We agree with the referee in considering the part on HF5 too long and distracting compared to the main aim of the work which was the critical discussion of microfluidic biosensors for the isolation and characterization of exosomes in the context of liquid biopsy. In particular, the work aims to present a critical review of the state of the art with the focusing on potential biosensors suitable for the simultaneous detection of intact exosomes and their cargos. This goal could allow the liquid biopsy to be performed and make it a routine approach for the early and specific diagnosis of cancer. We decided to add a brief experimental section which represent a first approach based on a promising microfluid tool (the field-flow fractionation) suitable for the intact exosome’s isolation from serum samples and integration with highly sensitive biosensing modules. According to your comments, we reduced and re-organize the section on HF5 results. In addition, we added comments to highlight advantages and disadvantage of the presented microfluid approaches. We also modified the Conclusions making them more relevant to the general issues critically discussed in the manuscript.
Below are my specific comments:
- A review article should discuss the current state and potential future advancements of a well-defined field or topic. Notably, a significant portion of section 4 is dedicated to presenting the working principles and results of a new biosensor.
We thank the referee, and we shorten and re-organized section 4.
- The conclusion reads more like a research article, focusing on the HF5 sensor, rather than summarizing insights from the field.
We thank the referee, and we modified the Conclusions to focalize it on general issues on microfluidic tools for exosomes isolation and cargo analysis.
- If the data is presented in an article, it should be laid out in a proper manner with methodology, materials, results, and discussion sections. Placing everything in the introduction is not appropriate.
See above comment, we modified section 4 including material and methods and results.
- Exosome isolation requires more rigorous validation. For example, In addition to the CD9 analysis, it is critical to perform protein analysis on the isolated exosomes to confirm the presence of other exosomal markers (such as TSG101, CD63 and CD81) and the absence of non-exosome markers (calnexin). Furthermore, the characterization of the isolated exosomes' size and morphology is noticeably absent.
We thank the referee. The experimental results presented are very preliminaries. The aim was to describe the HF5-based microfluidic systems as potential tool (proof of principle) for intact exosome isolation from blood plasma. Among intact exosome membrane markers, we used CD9 to explore the separative performances of HF5 system; not a complete validation was performed. We will consider analysis with other biomarkers for the validation of the new microfluidic biosensor. As for the size and morphological analysis, we used HF5-UV and MALS analysis to obtain data on fractionated particles, as discussed in the paragraph HF5 isolation of exosomes (Figure 7). The HF5-MALS has already shown able to give robust distribution analysis of dispersed nanosystems with a high efficiency and in native conditions. We will consider additional DLS and NTA analysis during the validation step of the new microfluidic biosensor.
- Figure 7-9 require revision. The clarity and image resolution of Figure 7 should be improved. Many elements within the figure are undersized and inadequately labeled, making it hard to understand. The figure legends in Figures 8 and 9 lack sufficient details.
We thank referee. Figure 7 (in the revised version Figure 8) was modified. Details were added to the legends of Figure 8 and 9 (in the revised version Figure 7 and 9).

Round 2
Reviewer 1 Report
Comments and Suggestions for Authors
Accept
Author Response
We thank Rewiever for the revison
Reviewer 2 Report
Comments and Suggestions for Authors
The manuscript can be accepted as a review paper.
Author Response
We thank Rewiever for the revison
Reviewer 3 Report
Comments and Suggestions for Authors
1. Was a statistical significance analysis conducted to determine the meaningfulness of the observed differences between HD and CRC samples in Figure 9.
2. There are errors in citations and referencing figures that need further clarification and revision . (For instance: Figure 5 II )
3. The line numbers are misaligned and occasionally obstruct the manuscript text and figures.
4. Some figures appear to be cropped out.
Author Response
The authors want to thank Reviewer for having revised our manuscript. All comments have been accepted and revision performed accordingly.
- Was a statistical significance analysis conducted to determine the meaningfulness of the observed differences between HD and CRC samples in Figure 9.
We thank the Reviewer. Details were added to the legend of Figure 9. “The graph shows the mean ± standard error of the mean (SEM) of the results for control (HD), and patients (CRC) samples (n = 6 for each condition). P-value < 0.05.”
- There are errors in citations and referencing figures that need further clarification and revision . (For instance: Figure 5 II ).
We have fixed the references and citations of the figures. (Figures 5, 7 and 8).
- The line numbers are misaligned and occasionally obstruct the manuscript text and figures.
Done.
- Some figures appear to be cropped out.
We thank the Reviewer; we adjusted the figures cropped out.